# *Brucella abortus* RNA does not polarize macrophages to a particular profile but interferes with M1 polarization

**Agustina Serafino, José L. Marin Franco, Mariano Maio, Aldana Trotta, Melanie Genoula, Luis A. Castillo, Federico Birnberg Weiss, José R. Pittaluga, Luciana Balboa, Paula Barrionuevo⬡*[ID], M. Ayelén Milillo[ID]*⬡**

Instituto de Medicina Experimental—Consejo Nacional de Investigaciones Científicas y Técnicas (CONICET), Academia Nacional de Medicina; Buenos Aires, Argentina

⬡ These authors contributed equally to this work.
* pbarrion2004@yahoo.com.ar (PB); mililloayelen@gmail.com (MAM)

**Data Availability Statement:** All relevant data are within the manuscript and its Supporting Information files.

## Abstract

Monocytes and macrophages play a central role in chronic brucellosis. *Brucella abortus* (*Ba*) is an intracellular pathogen that survives inside these cells. On the other hand, macrophages could be differentiated into classical (M1), alternative (M2) or other less-identified profiles. We have previously shown that *Ba* RNA (a bacterial viability-associated PAMP or *vita*-PAMP) is a key molecule by which *Ba* can evade the host immune response. However, we did not know if macrophages could be polarized by this *vita*-PAMP. To assess this, we used two different approaches: we evaluated if *Ba* RNA *per se* was able to differentiate macrophages to M1 or M2 or, given that *Ba* survives inside macrophages once a Th1 response is established (*i.e.*, in the presence of IFN-γ), we also analysed if *Ba* RNA could interfere with M1 polarization. We found that *Ba* RNA alone does not polarize to M1 or M2 but activates human macrophages instead. However, our results show that *Ba* RNA does interfere with M1 polarization while they are being differentiated. This *vita*-PAMP diminished the M1-induced CD64, and MHC-II surface expression on macrophages at 48 h. This phenomenon was not associated with an alternative activation of these cells (M2), as shown by unchanged CD206, DC-SIGN and CD163 surface expression. When evaluating glucose metabolism, we found that *Ba* RNA did not modify M1 glucose consumption or lactate production. However, production of Nitrogen Reactive Species (NRS) did diminish in *Ba* RNA-treated M1 macrophages. Overall, our results show that *Ba* RNA could alter the proper immune response set to counterattack the bacteria that could persist in the host establishing a chronic infection.

## Author summary

Brucellosis is a zoonotic disease, caused by bacteria from *Brucella* genus. It is distributed worldwide. The disease affects male and female livestock fertility and causes abortions. Despite eradication programmes, brucellosis continues to be a big issue for public health

**Funding:** This work was supported by two grants which belong to PB: Proyectos de Investigación Científica y Tecnológica (PICTs) numbers 2016-0356 and 2018-1969 funded by the Agencia Nacional de Promoción Científica y Tecnológica (ANPCYT-Argentina). The funders had no role in study design, data collection and analysis, decision to publish, or preparation of the manuscript.

**Competing interests:** The authors have declared that no competing interests exist.

and for economy policies. Hence, to design mitigation strategies and diminish economic burden, we need to understand the pathogenesis of this disease in humans and livestock. A vexing enigma of *Brucella* immunity resides in the long-term persistence of the infecting bacterium despite a vigorous and specific immune response. This manuscript shed light into a relevant aspect whereby *Ba* evades immune host surveillance: the interference with M1 macrophage polarization, phenomenon mediated by *Ba* RNA. Our findings demonstrate that *Ba* RNA constitute a novel virulence factor whereby this bacterium interferes with M1 polarization of human macrophages. These results provide new insights about how *Ba* can persist in the presence of Th1 responses, *i.e.*, in the presence of IFN-γ establishing a chronic infection.

## Introduction

Macrophages play a key role in immunity responses against pathogens. Not only do these cells have phagocytic functions but also discriminate self from non-self antigens. Once outside the bone marrow, monocytes can migrate to different tissues and differentiate into macrophages exhibiting a wide array of phenotypic and functional characteristics. Initially, two macrophage subpopulations had been described: classical macrophages (or M1; CD14$^{++}$CD16$^-$) and alternative macrophages (or M2; CD14$^+$CD16$^{++}$). More recently, other profiles have been incorporated, such as tumour-associated macrophages (or TAM), CD169$^+$ macrophages, which are implicated in immune tolerance and antigen presentation and TCR$^+$ macrophages [1,2]. M1 macrophages secrete pro-inflammatory cytokines, mediate resistance against pathogens and show potent microbicidal properties. However, they also contribute to the destruction of tissues. M2 macrophages are essential in parasitic infections, tissue remodelling, allergies, and angiogenesis [1,2].

Brucellosis is an infectious disease driven by bacteria from genus *Brucella*. *Brucella abortus* (*Ba*) is an intracellular pathogen whose preferential niches to persist are monocytes, macrophages, and trophoblastic cells. The infection with *Ba* triggers the innate and adaptive immunity towards Th1 and activation of CD8$^+$ T cells [3–5]. Despite eliciting an immune response, this bacterium relies on several strategies to persist inside cells, evading host defence. Previous results from our laboratory demonstrate that infection of monocytes/macrophages with *Ba* diminishes MHC-I and MHC-II IFN-γ-induced surface expressions, impairing antigen presentation to T cells [6–9]. Moreover, a pathogen-associated molecular pattern (PAMP) associated with bacterial viability (*vita*-PAMP), the *Ba* RNA, sensed by Toll like receptors (TLR) 7/8 [10] is involved in both phenomena [11,12]. MHC-I and II down-modulation cause a reduction in antigen presentation of RNA-treated macrophages to CD8$^+$ and CD4$^+$ T cells, respectively [11,12]. However, we could not determine whether macrophages are differentiated to a particular profile or whether other macrophage functions were also affected. In line with this, Saha *et al.* (2017) demonstrated that TLR7/8 signalling triggered by Hepatitis C virus (HCV) promotes the differentiation of monocytes and polarization of macrophages towards an M2 profile [13]. We also demonstrated that RNA from other microorganisms (*Escherichia coli*, *Klebsiella pneumoniae*, *Trypanosoma cruzi* and *Salmonella typhimurium*) is also able to diminish MHC-I and II IFN-γ-induced surface expression on human monocytes [11,12]. Moreover, other pathogens activate macrophages via recognition of their RNA. The micro RNA 3 encoded by the Hepatitis B virus (HBV-miR-3) triggers M1 polarization on macrophages [14]. *Streptococcus pyogenes* RNA is sensed by TLR13 in murine bone marrow-derived macrophages and induce an immune response *in vitro* [15].

Regarding metabolic pathways in macrophages, M1 cells oxidise glucose towards aerobic glycolysis and lactate production (rapid fuel of energy) while M2 shifts the glucose

consumption towards tricarboxylic acid and oxidative phosphorylation [16–18]. So, we also wondered if *Ba* RNA was also interfering with metabolic pathways on macrophages.

Hence, considering the evidence and the events that occur on other infectious processes, we hypothesized that *Ba* RNA favours the differentiation of macrophages to an M1 profile at early time points, but it interferes with M1 profile and/or change towards M2 at later time points. To test our hypothesis, we aimed to determine the effect of *Ba* RNA on macrophage polarization *per se* or during the M1 differentiation program.

## Methods

### Ethics statement

Human monocytes were isolated exclusively from healthy adult donors according to the rules of the Ethics Committee of the Instituto de Medicina Experimental (IMEX)—Consejo Nacional de Investigaciones Científicas y Técnicas (CONICET), Academia Nacional de Medicina. All donors signed an informed consent prior to the study. The donation was anonymous, and all the information of the donors is protected. Bone marrow derived murine macrophages (BMM) were generated by differentiation of bone marrow progenitors from female C57BL/6 mice (age: 2–3 months). All procedures were approved and carried out under the conditions of the Institutional Committee on the Care and Use of Laboratory Animals (CICUAL: Comité Institucional para el Cuidado y Uso de Animales de Laboratorio) of the IMEX.

### Bacteria strains

*Brucella abortus* S2308, was grown on triptein soy agar medium, supplemented with yeast extract (Merck Millipore). The number of bacteria in the stationary phase cultures (in triptein soy broth) was determined by comparing the optical density at 600 nm with a standard curve. The cultures in liquid media of *Ba* were carried out in the Biosafety Level 3 laboratory (BSL3) of the Operational Unit Center for Biological Containment (UOCCB) of the National Administration of Laboratories and Health Institutes 'Dr. Carlos G. Malbrán' (ANLIS-Malbrán), thanks to an agreement established in 2017.

### Cell cultures

Peripheral blood mononuclear cells (PBMCs) were obtained by centrifugation in Ficoll-Hypaque gradients (GE Healthcare) from human blood collected from healthy adult donors. Human monocytes were obtained after centrifugation of the PBMCs in a Percoll gradient (GE Healthcare) and then resuspended in standard medium. The purity of the isolated CD14+ monocytes was greater than 80%, determined by flow cytometry. Cell viability was measured by the trypan blue exclusion test and was greater than 95% in all experiments. BMMs were generated by differentiation of bone marrow progenitors obtained from C57BL/6 mice, as previously described [19].

All the experiments were carried out in an incubator at 37°C in an atmosphere with 5% $CO_2$. The standard medium used was composed of RPMI 1640 supplemented with 25mM Hepes, 2mM L-glutamine, 10% heat-inactivated fetal bovine serum (FBS, Gibco), 100U of penicillin.ml$^{-1}$ and 100 μg of streptomycin.ml$^{-1}$.

### RNA preparation

5 x 10$^9$ CFU of *Ba* were resuspended in 1 ml of TRIzol (Invitrogen) and their RNA was purified using Quick-RNA DirectZol columns (Zymo Research) according to the manufacturer's instructions. The quantification and purity of the *Ba* RNA was determined using a DeNovix

DS-11 spectrophotometer (DeNovix Inc.). In all cases, the absorbance ratio 260/280 was higher than 2.0 and the ratio 260/230 was higher than 1.8.

### Viability assay

For viability assays, human peripheral blood (PB) derived macrophages were treated with *Ba* RNA in the presence or absence of IFN-γ and LPS for 24 and 48 h. Cells treated with 2% paraformaldehyde (PFA) were included as a positive control of the technique. At both times, cells were washed and incubated with Annexin V-FITC and Propidium Iodide (IP) (BD Biosciences) for 10 minutes on ice and in the dark. After staining, cells were analysed with FACSCalibur (BD Biosciences) or Partec CyFlow (LabSystems) flow cytometers. Cells were evaluated in the Annexin V$^+$/IP$^-$ (early apoptosis), Annexin V$^+$/IP$^+$ (late apoptosis) and Annexin V$^-$/IP$^+$ (necrosis) quadrants. Data analysis was performed with FlowJo 7.6.2 software (FlowJo, LLC).

### *In vitro* stimulation

In experiments with murine macrophages, BMMs were treated with *Ba* RNA in the presence of 10 ng/ml of recombinant murine IFN-γ (mIFN-γ, PeProtech) and *E. coli* LPS for different times. In experiments with human macrophages, 2.5 x 10$^5$ PB-human monocytes were stimulated with GM-CSF for 5–7 days prior to stimulation with *Ba* RNA in the presence or absence of IFN-γ and *E. coli* LPS. Cell cultures were incubated for 48 h at 37˚C in a 5% CO$_2$ atmosphere. In all cases, the expression of the different cell markers was evaluated by flow cytometry, as described in the following paragraph.

### Flow cytometry

Macrophages from PB-purified monocytes were stained with the anti-human MHC-II conjugated to PE (clone L243, BD) or anti-human CD86 conjugated to PE (clone IT2.2; BD) or anti-human CD64 conjugated to FITC (clone 10.1; BD) or anti-human CD206 conjugated to PE (clone 19.2, BD) or anti-human DC-SIGN conjugated to FITC (clone DCN46, BD) or anti-human CD163 conjugated to PerCP (clone GHI61, Biolegend) or anti-mouse MHC-II conjugated to FITC (I-A/I-E, clone M5/114.15.2, e-Bioscience) or the corresponding isotype control antibodies and then evaluated by flow cytometry at different times post-stimulation. When possible, cells were washed and incubated with 7-AAD (BD Biosciences) for 10 minutes on ice and in the dark. The expression of the different surface markers was evaluated within the viable cell population (7-AAD negative). After labelling, the cells were analysed on a FACSCalibur flow cytometer (BD Biosciences) or Partec CyFlow (LabSystems) and the data were processed with the FlowJo 7.6.2 or vX.0.7 applications (FlowJo, LLC). Data was normalized to untreated cells, *i.e.*, in each set of experiments, for each marker, the MFI of treatments (from the % of positive cells for each marker) was divided by MFI of media condition (from the % of positive cells for each marker).

### Determination of cytokine concentration

The concentration of different cytokines and chemokines corresponding to the M1 profile (TNF-α, IL-1β and IL-8) and M2 (IL-10) was quantified in supernatants (SNs) from human macrophages stimulated with *Ba* RNA for 24 and 48 h by ELISA sandwich, according to the manufacturer's instructions. The following detection kits were used: TNF-α (e-Bioscience), IL-1β (e-Bioscience), IL-8 (BioLegend) and IL-10 (BioLegend).

### Determination of Nitrogen Reactive Species (NRS)

SNs were harvested from *Ba* RNA-treated BMMs in the presence of mIFN-γ and LPS for 24 and 48 h. The content of NRS was measured by the determination of nitrites with the Griess reagent (Reagent A: sulfanilic acid, Reagent B: α-naphthylamine) by comparison with a standard curve of OD at 550 nm, as described in [20]. The results were expressed as micromoles of nitrite.

The determination of the inducible NO synthase was done as described before [21]. Briefly, treated BMMs were washed with PBS, fixed with 0.05% paraformaldehyde. Then cells were treated with 0.25% Triton X-100 for 15 min, blocked with 3% bovine serum albumin (BSA) for 1 hr and stained with an anti-iNOS antibody (clone 6, BD Biosciences) for 30 min. An isotype-matched antibody was assayed in parallel, and fluorescence was determined by BD FACSCalibur on 10,000 events per sample.

### Lactate production and glucose concentration

Lactate production and glucose concentrations in the culture medium was measured using the spectrophotometric assays Lactate Kit and Glicemia Enzimatica AA Kit both from Wiener (Argentina) as previously described [18]. Basically, the kits are based on the oxidation of lactate or glucose, respectively, and the subsequent production of hydrogen peroxide. The consumption of glucose was determined by assessing the diminution of glucose levels in culture supernatants in comparison with RPMI 10% FBS. The absorbance was read using a Biochrom Asys UVM 340 Microplate Reader and software.

### Statistical analysis

Results were analysed with one-way ANOVA followed by *post hoc* Tukey test or two-way ANOVA followed by *post hoc* Bonferroni test with GraphPad Prism software, 8.4.3 version.

## Results

### *Ba* RNA activates but does not polarize macrophages towards M1

We first evaluated whether *Ba* RNA *per se* was able to polarize human macrophages towards M1. Human monocytes were differentiated to macrophages with GM-CSF for 5–7 days and then stimulated with *Ba* RNA (5 μg/ml) for 24 or 48 h. MHC-II, CD86 and CD64 (M1 markers) expression was then assessed by flow cytometry. (IFN-γ + LPS)-treated macrophages were used as a positive control of M1 macrophages. *Ba* RNA did not significantly modify the surface expression of MHC-II and CD86 (Fig 1A and 1B) neither at 24 nor 48 h. However, there was a reduction in CD64 expression in *Ba* RNA-treated macrophages at 48 h (Fig 1C). Supernatants at each time were harvested and IL-1β, TNF-α and IL-8 were quantified by ELISA sandwich. *Ba* RNA induced the secretion of all pro-inflammatory cytokines by macrophages, except for IL-8, at 24 h but not at 48 h (Fig 2A–2C).

We also performed an Annexin assay to evaluate whether there was a loss of cell viability in *Ba* RNA stimulated cultures. This treatment did not induce early and late apoptosis or necrosis (S1 Fig).

These results indicate that *Ba* RNA activates -temporarily though- macrophage immune responses but is not able to polarize *per se* those cells to M1 macrophages.

### *Ba* RNA activates but does not polarize macrophages towards M2

Next, we evaluated whether *Ba* RNA was able to polarize macrophages towards M2. We performed the same experiments as described before, and CD206, DC-SIGN and CD163

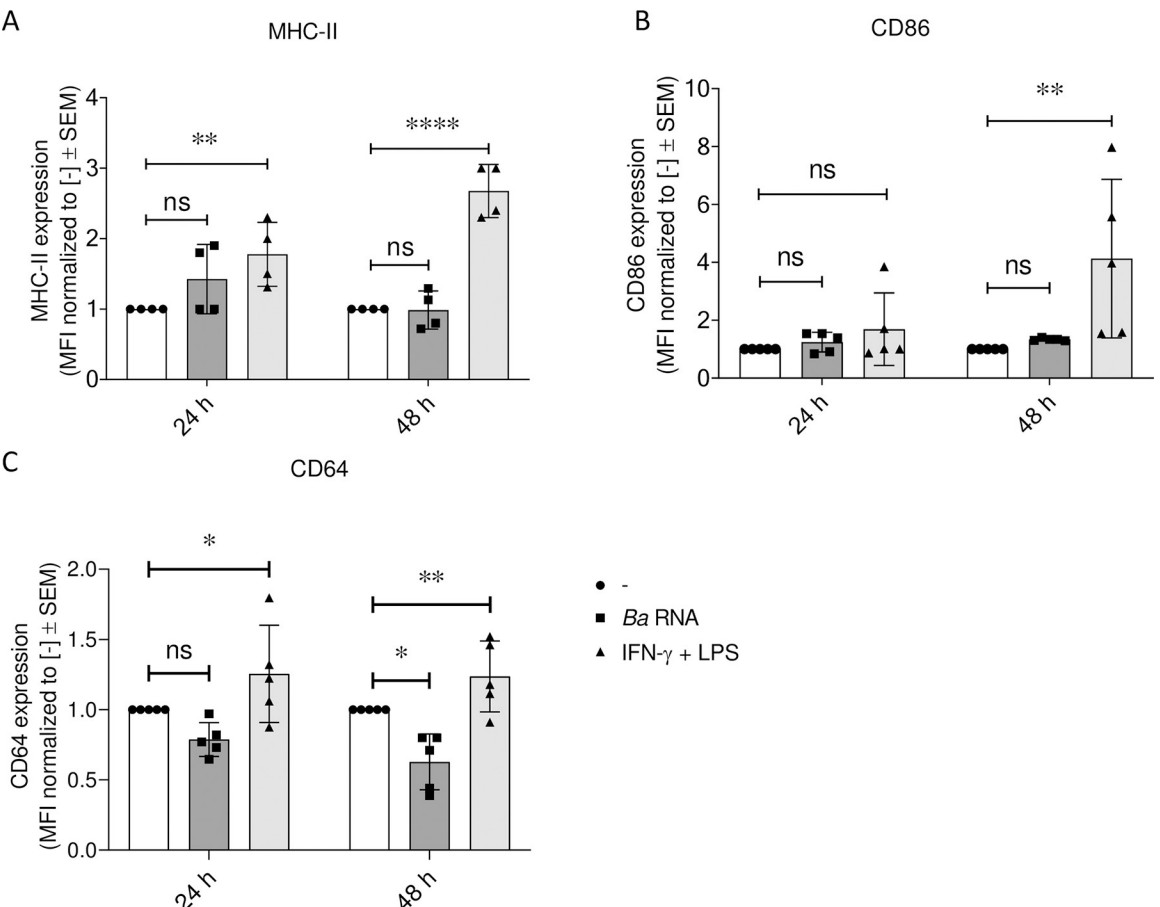

**Fig 1. *Ba* RNA interferes with the expression of M1 markers at 48 h.** Human M0 macrophages were stimulated with *Ba* RNA for 24 and 48 h. (A) MHC-II, (B) CD86 and (C) CD64 (M1 markers) expressions were determined by flow cytometry. Cells treated with IFN-γ and LPS were used as a positive control. Bars indicate the geometric means ± normalized to untreated cells ± SEM of at least four independent experiments. MFI, mean fluorescence intensity. ns, non-significant; *P<0.05; **P<0.01; ****P<0.0001 vs. untreated cells (-).

expressions were measured as typical M2 markers. IL-4-treated macrophages were used as a positive control for CD206 and DC-SIGN expressions and IL-10-treated macrophages for CD163. No significant differences among DC-SIGN or CD163 were observed in *Ba* RNA-treated macrophages in comparison with untreated macrophages at both times (Fig 3B and 3C). On the contrary, CD206 expression was reduced in *Ba* RNA-treated macrophages at both times (Fig 3A).

Supernatants were harvested and the anti-inflammatory cytokine IL-10 was quantified. *Ba* RNA induced the secretion of this cytokine at 24 and 48 h (Fig 4). Hence, this PAMP can activate macrophages but does not polarize these cells towards a define profile.

## *Ba* RNA interferes with the expression of pro-inflammatory markers during M1 macrophage polarization

We showed that *Ba* RNA does not polarize macrophages *per se*. It is already elucidated that *Ba* infection activates Th1 cells with the consequent secretion of IFN-γ and induction of CD8[+] T cells. Moreover, we have previously demonstrated that *Ba* RNA diminishes the IFN-γ and LPS-induced MHC-II surface expression on human monocytes altering antigen presentation

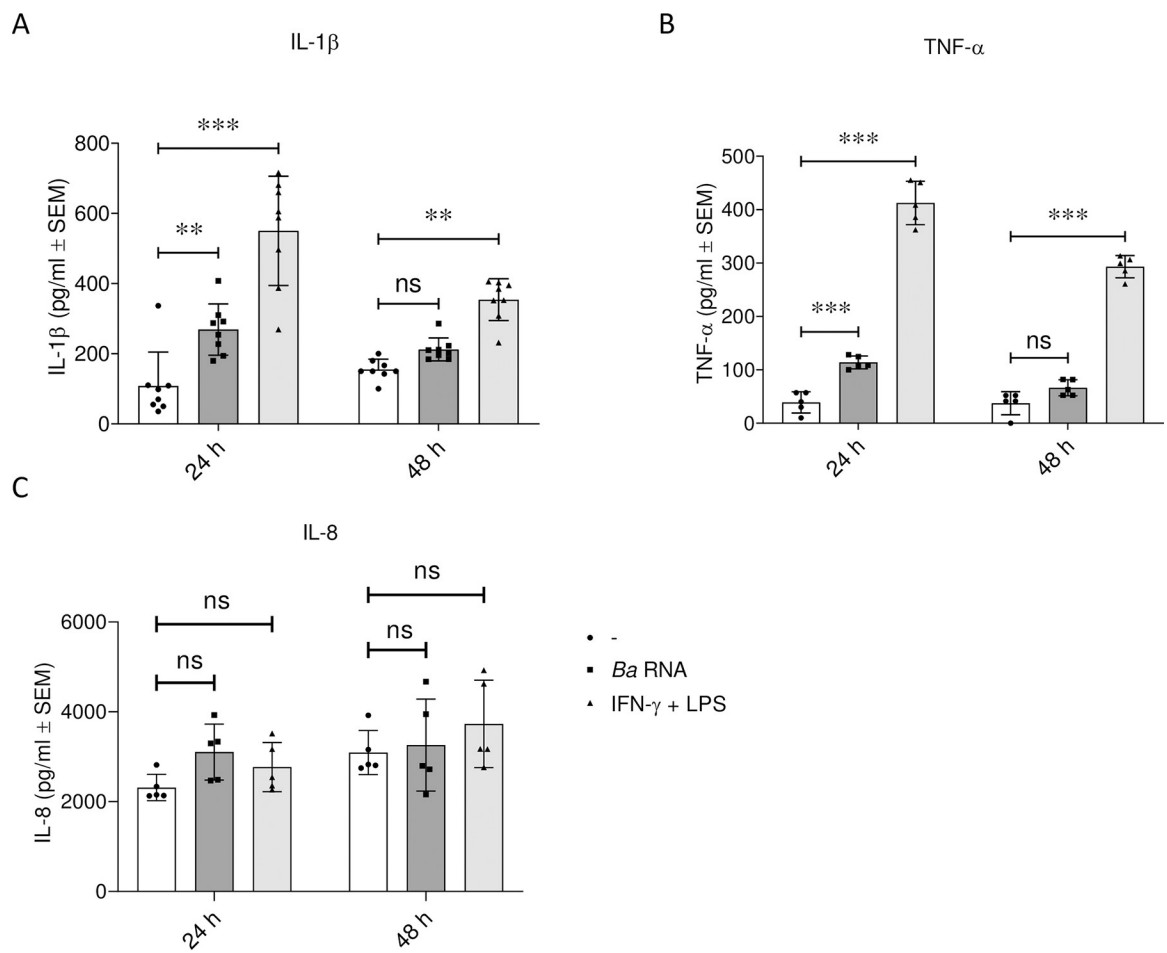

**Fig 2. *Ba* RNA induces the secretion of pro-inflammatory cytokines at 24 h.** Human M0 macrophages were stimulated with *Ba* RNA for 24 and 48 h. Secretion of (A) IL-1β, (B) TNF-α and (C) IL-8 was quantified by ELISA sandwich. Cells treated with IFN-γ and LPS were used as a positive control. Bars indicate the geometric means ± SEM of at least five independent experiments. ns, non-significant; **P<0.01; ***P<0.001 vs. untreated cells (-).

to CD4$^+$ T cells [12]. So, we wondered whether *Ba* RNA affects M1 polarization while it is occurring, *i.e.*, in macrophages cultivated in the presence of IFN-γ and LPS.

To evaluate whether *Ba* RNA was able to modify macrophage inflammatory profile while they are being polarized to M1, human monocytes were differentiated to macrophages as mentioned above and then stimulated with *Ba* RNA (5 μg/ml) in the presence of IFN-γ + LPS for 24 and 48 h. The expression of MHC-II, CD86 and CD64 was assessed by flow cytometry. At 24 h there were no differences in the expression of MHC-II and CD86 in cells treated with *Ba* RNA + (IFN-γ and LPS). CD64 expression was significantly reduced though (Fig 5C). At 48 h there was a significant reduction in the expression of MHC-II and CD64 in *Ba* RNA + (IFN-γ and LPS)-treated macrophages (Fig 5A and 5C). There were no changes in CD86 expression at any evaluated time points in comparison with IFN-γ and LPS-treated cells (Fig 5B).

We also performed an Annexin assay to evaluate whether there was a loss of cell viability in *Ba* RNA + (IFN-γ + LPS)-stimulated cultures. This treatment did not induce early or late apoptosis or necrosis (S2 Fig).

Moreover, we determined the ability of *Ba* RNA to induce the secretion of pro-inflammatory cytokines from macrophages during M1 polarization. IL-1β, TNF-α and IL-8 were

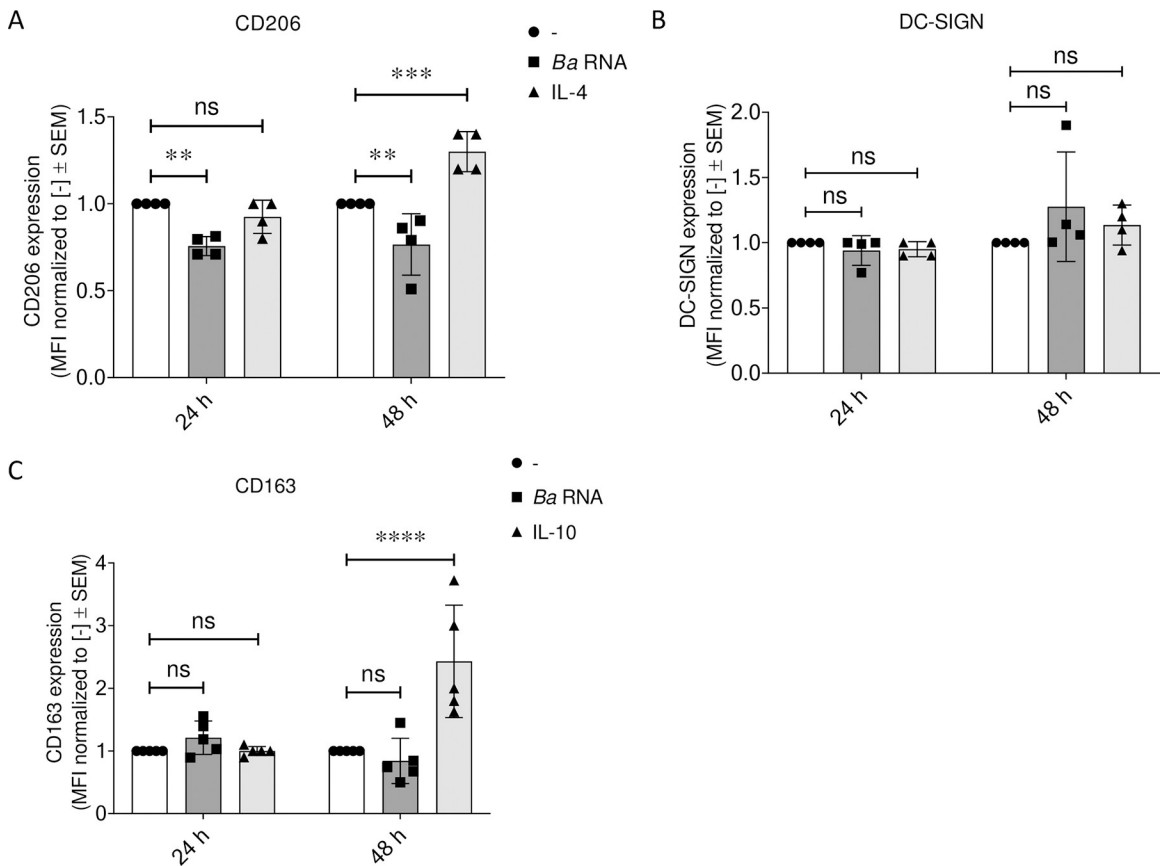

**Fig 3. *Ba* RNA does not modulate the expression of M2 markers.** Human M0 macrophages were stimulated with *Ba* RNA for 24 and 48 h. (A) CD206, (B) DC-SIGN and (C) CD163 (M2 markers) expressions were determined by flow cytometry. Cells treated with IL-10 or IL-4 were used as controls. Bars indicate the geometric means normalized to untreated cells ± SEM of at least four independent experiments. MFI, mean fluorescence intensity. ns, non-significant; **P<0.01; ***P<0.001; ****P<0.0001 vs. untreated cells (-).

quantified in the supernatants of *Ba* RNA plus IFN-γ and LPS-treated macrophages at 24 and 48 h post-stimulation. There were no significant changes in the secretion of cytokines at any evaluated time point (Fig 6). We also calculated the ratio of pro and anti-inflammatory cytokines. At 48 h, there was a reduction tendency -non-significant though- in the ratio TNF-α/IL-10 in *Ba* RNA-treated macrophages during M1 polarization (S3 Fig), implying that there could be a predominant anti-inflammatory cytokine context at later times post-stimulation.

As consequence, at least phenotypically, *Ba* RNA interferes with M1 polarization.

## *Ba* RNA does not modulate M2 markers under M1 polarizing conditions

Since *Ba* RNA interferes with M1 polarization, we asked whether this event was accompanied with a differentiation to an M2 profile. To evaluate this, the expression of M2 markers (CD163, CD206 and DC-SIGN) was determined during M1 polarizing conditions at 24 and 48 h post-stimulation.

Also, in culture supernatants we quantified the secretion of IL-10. As observed, there were neither no significant changes on CD206 and DC-SIGN surface expression nor in the IL-10 secretion mediated by *Ba* RNA in the presence of IFN-γ and LPS. However, there was a transient induction of CD163 surface expression on *Ba* RNA-treated macrophages in the presence of IFN-γ and LPS at 24 h (Fig 7A–7D).

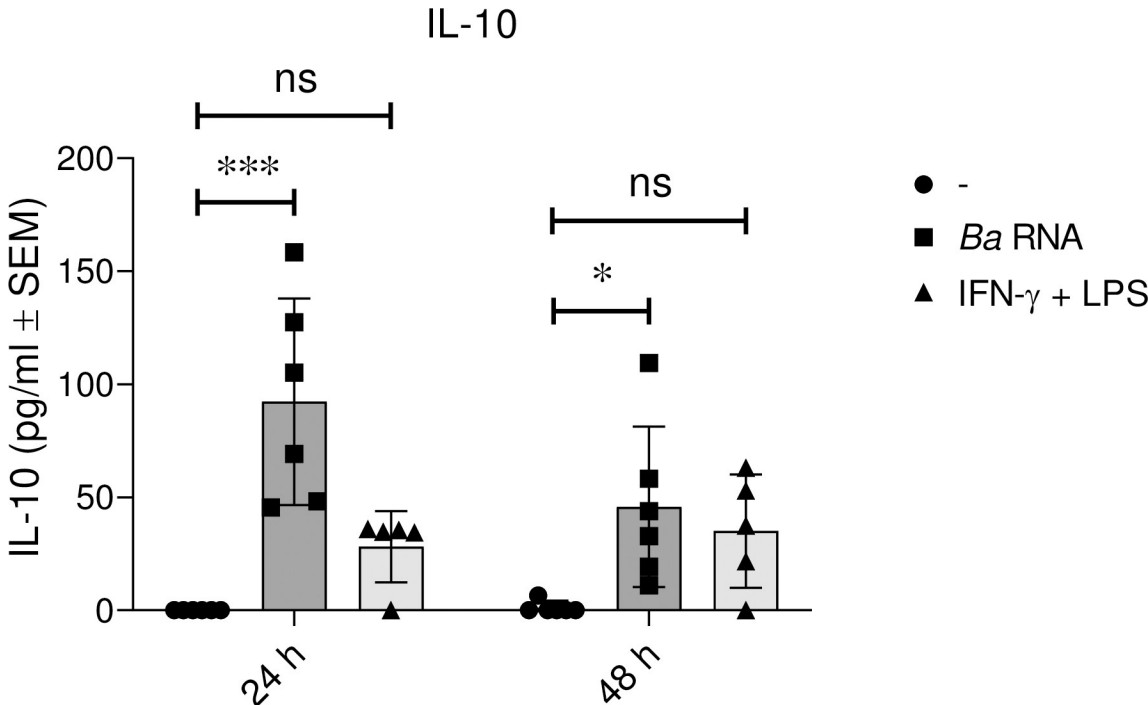

**Fig 4. *Ba* RNA induces IL-10 secretion.** Human M0 macrophages were stimulated with *Ba* RNA for 24 and 48 h. Secretion of IL-10 was quantified by ELISA sandwich in supernatants. Cells treated with IFN-γ and LPS were used as a positive control. Bars indicate the geometric means ± SEM of six independent experiments. ns, non-significant; *P<0.05; ***P<0.001 vs. untreated cells (-).

So, these results altogether show that *Ba* RNA interferes with M1 polarization, but not because of macrophages turning into an M2 profile.

### *Ba* RNA induces glucose consumption at 24 h and lactate production at 48 h

Given the fact that metabolic features of M1 macrophages are increased glucose consumption to produce lactate and diminished OXPHOS metabolic pathway, we determined the glucose consumption of *Ba* RNA or *Ba* RNA + (IFN-γ + LPS)-treated macrophages at 24 and 48 h post-stimulation. As demonstrated in Fig 8A, *Ba* RNA induced glucose consumption at 24 h and lactate production at 48 h, confirming its pro-inflammatory properties, at least transiently. On the contrary, *Ba* RNA + (IFN-γ + LPS)-treated macrophages did not change glucose consumption and lactate production compared to (IFN-γ + LPS)-treated macrophages (Fig 8A and 8B). Hence, this phenotypic pathway of M1 metabolism is not altered by *Ba* RNA.

### *Ba* RNA diminishes the production of Nitrogen Reactive Species (NRS) during M1 polarization at 48 h

Since we previously showed that *Ba* RNA diminishes the antigen presentation capacity of macrophages to CD8[+] and CD4[+] T cells [11,12], we wondered if microbicidal capacity (as a key feature of M1 profile) of *Ba* RNA-treated macrophages in the presence of IFN-γ and LPS was altered as well. So, we performed two types of experiments: Griess assay and intracellular iNOS expression. To perform these experiments, we used murine BMM. Firstly, we evaluated whether *Ba* RNA was able to interfere with M1 polarization in BMM. *Ba* RNA diminished the (mIFN-γ + LPS)-induced MHC-II surface expression at 48 h (S4 Fig). Then, we determined

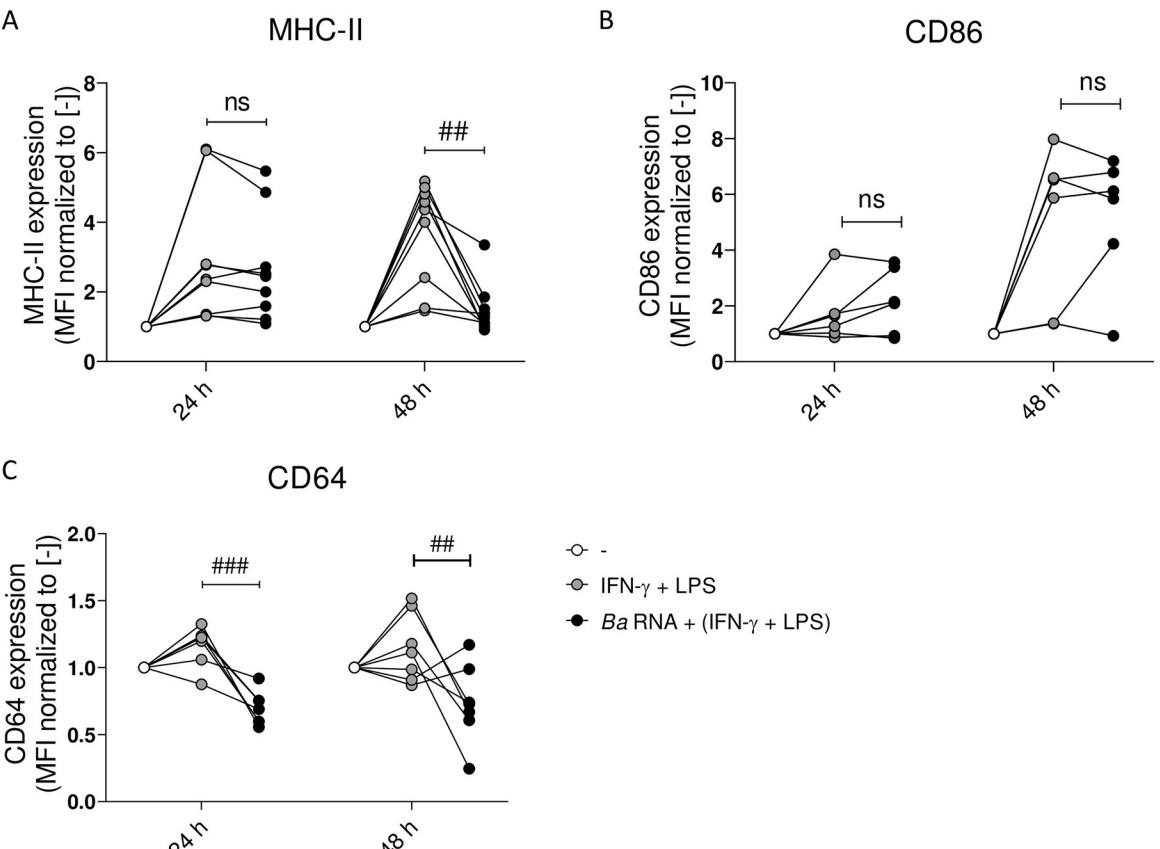

**Fig 5. *Ba* RNA interferes with the expression of M1 markers at 48 h.** Human M0 macrophages were stimulated with *Ba* RNA for 24 and 48 h in the presence of IFN-γ + LPS. (A) MHC-II, (B) CD86 and (C) CD64 (M1 markers) expressions were determined by flow cytometry. Cells treated with IFN-γ and LPS were used as a positive control. Dots indicate the geometric means normalized to untreated cells of at least six independent experiments. MFI, mean fluorescence intensity. ns, non-significant; ##P<0.01; ###P<0.001 vs. cells treated with IFN-γ + LPS.

NRS in culture supernatants of *Ba* RNA-treated BMM during M1 polarization and the intracellular expression of iNOS.

As observed, at 24 h, *Ba* RNA-treated BMMs in the presence of mIFN-γ and LPS did not induce or reduce the secretion of NRS (detected as nitrite oxide in supernatants) or the expression of iNOS. However, at 48 h NRS secretion diminished in BMMs treated with *Ba* RNA in the presence of mIFN-γ and LPS (Fig 9A). Although not significant, this could be related with a decreased iNOS expression at least at 48 h (Fig 9B).

## Discussion

Brucellosis is a zoonotic disease, caused by bacteria from *Brucella* genus. It is distributed worldwide, especially affecting livestock from Africa, Latin America, New Zealand and Australia, India, Saudi Arabia, and Mediterranean countries among others. The main pathogenic species for livestock are *B. abortus*, responsible for bovine brucellosis, *B. melitensis*, the main aetiological agent of brucellosis in both small ruminants and humans, *B. suis*, responsible for swine brucellosis and *B. ovis*, responsible for sheep brucellosis. The disease affects male and female livestock's fertility and causes abortions. Despite eradication programmes, brucellosis continues to be a big issue for public health and for economy policies due to the loss of money

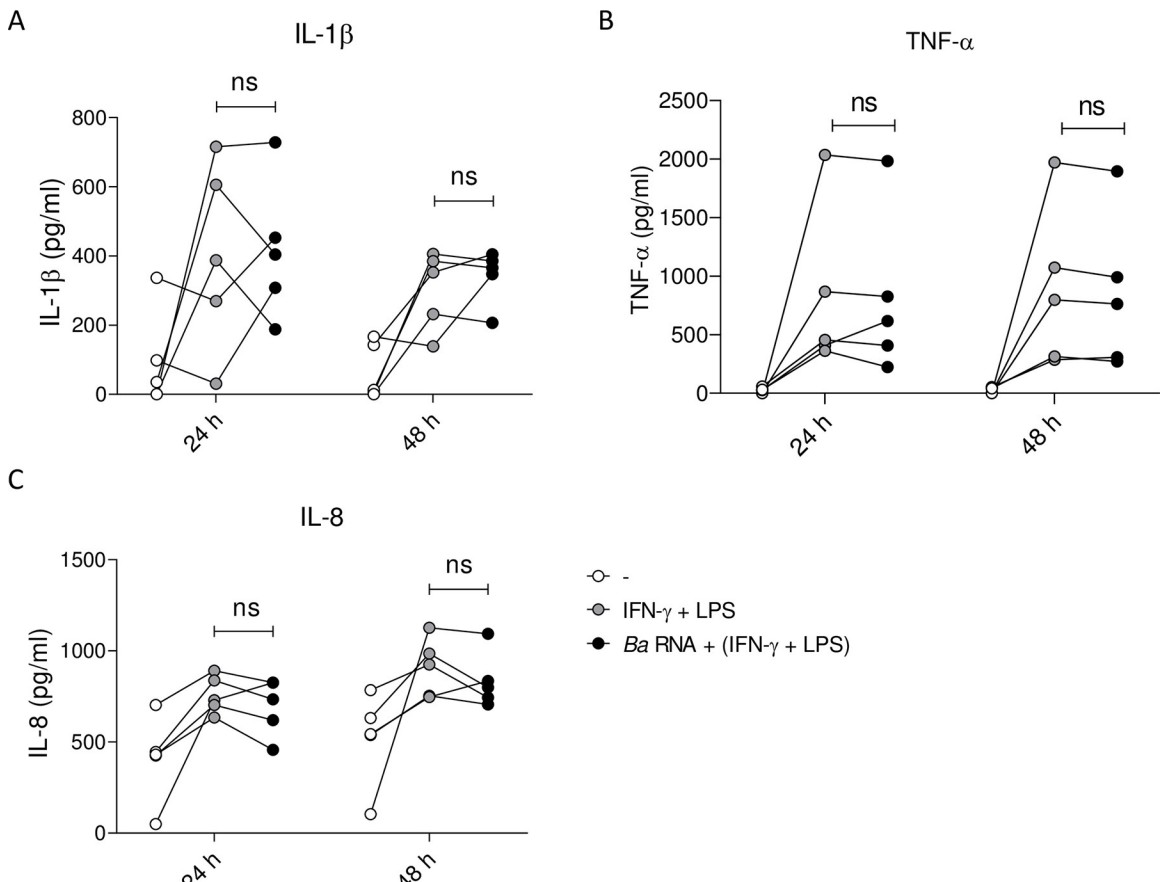

**Fig 6. *Ba* RNA does not interfere with the secretion of pro-inflammatory cytokines.** Human M0 macrophages were stimulated with *Ba* RNA for 24 and 48 h in the presence of IFN-γ + LPS. Secretion of (A) IL-1β, (B) TNF-α and (C) IL-8 was quantified by ELISA sandwich. Cells treated with IFN-γ and LPS were used as a positive control. Dots indicate the geometric means of at least five independent experiments. ns, non-significant.

driven by the need of 'brucellosis free'-status livestock. Hence, to design mitigation strategies and diminish economic burden, there is a common need of understanding the pathogenesis of this disease in humans and livestock.

*Brucella* spp., like most intracellular bacteria, display strategies to evade or counterattack the host immune responses. By achieving this, a chronic infection could be established. These bacteria can live intracellularly in macrophages, their quintessential replicative niche [22].

Monocytes and macrophages play a key role against infections mainly through phagocytosis and antigen presentation [23]. Peripheral blood monocytes migrate inside the body tissues and polarize to macrophages of different subclasses depending on the microenvironment. Among the subclasses of macrophages, M1 (formerly known as 'classical macrophages') and M2 (formerly known as 'alternative macrophages') are the two main subpopulations. IFN-γ and LPS differentiate unpolarized (M0) macrophages to M1 while IL-4 and IL-13, IL-10 or corticosteroids differentiate M0 to M2. Basically, M1 cells eliminate pathogens but also damage the tissues while M2 cells suppress inflammation and mediate tissue healing. Both populations are necessary to solve the infectious process [2].

What is particularly interesting for *Ba* is that monocytes and macrophages constitute the first line of defense against the bacteria but also are the place where they reside [24].

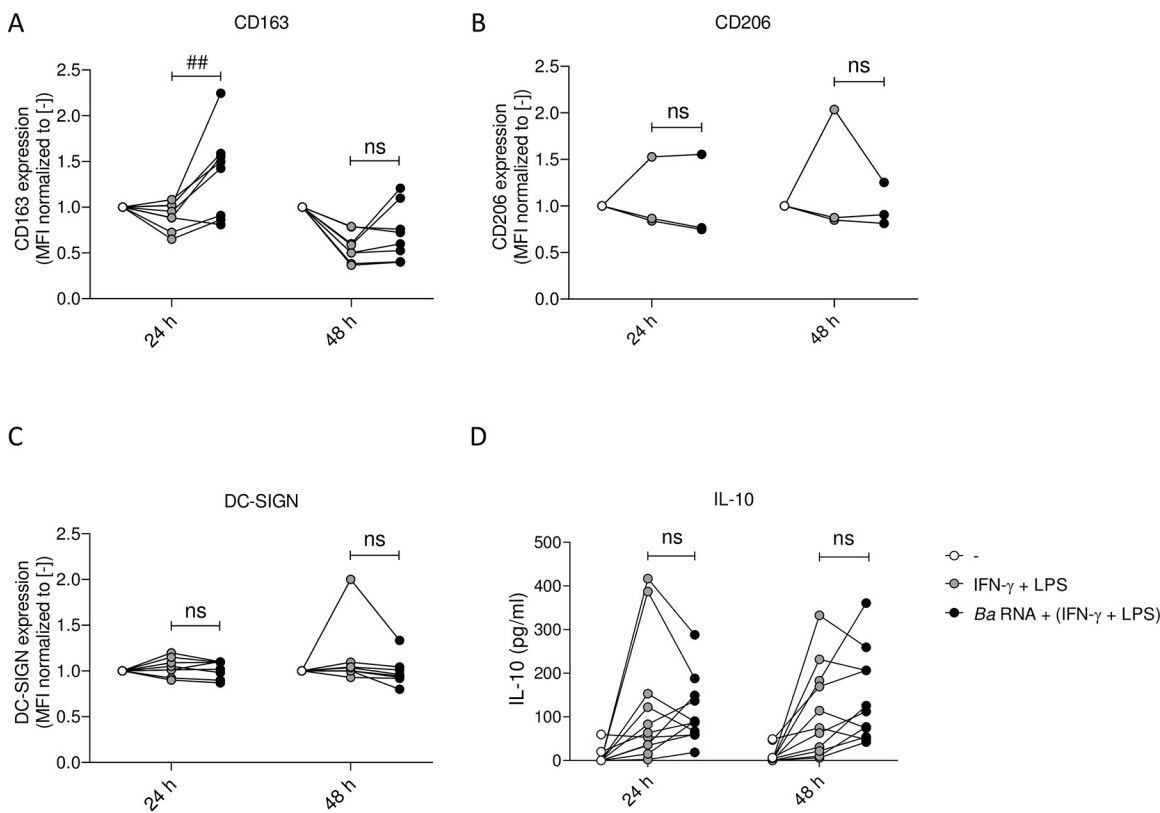

**Fig 7. *Ba* RNA does not modulate the expression of M2 markers under M1 polarizing conditions.** Human M0 macrophages were stimulated with *Ba* RNA for 24 and 48 h in the presence of IFN-γ + LPS. (A) CD163, (B) CD206 and (C) DC-SIGN (M2 markers) expressions were determined by flow cytometry. Dots indicate the geometric means normalized to untreated cells of at least three independent experiments. (D) Secretion of IL-10 was quantified by ELISA sandwich in supernatants. Dots indicate the geometric means of eleven independent experiments. Cells treated with IFN-γ and LPS were used as a positive control. MFI, mean fluorescence intensity ns, non-significant; ##P<0.01 vs. cells treated with IFN-γ + LPS.

We and others have demonstrated that infection of monocytes/macrophages with *Ba* provokes an alteration of various functions of those cells: 1) reduced IFN-γ-induced expression of MHC-I and MHC-II and consequent impaired antigen presentation capacity [6–9]; 2) diminished phagocytosis [25]; 3) interfered TLRs signalling [26,27] and 4) inhibited apoptosis [28,29]. Altogether, these impaired tasks of monocytes/macrophages led to immune evasion and survival of *Ba* inside the host.

Results from our group have shown that a *Ba* PAMP associated with bacterial viability (*vita*-PAMP) is responsible for the IFN-γ-induced MHC-I and MHC-II down-modulation on monocytes/macrophages. This *vita*-PAMP is *Ba* RNA [11,12]. We have shown that Toll-like receptor 8 (TLR8) is, at least, one receptor by which *Ba* RNA is sensed [11]. Besides, stimulation of human monocytes by *Ba* RNA or synthetic TLR8 ligands promotes the early secretion of proinflammatory mediators (TNF-α e IL-1β) [11]. All these results and the importance of monocytes/macrophages in the resolution of *Ba* infection laid the ground for this work. Therefore, we aimed to study the role of *Ba* RNA during the polarization of macrophages.

Considering that infection with *Ba* starts with the activation of innate immunity [30] we wondered whether *Ba* RNA *per se* could constitute enough stimulus for macrophages to be polarized to M1 or M2. Our results show that *Ba* RNA can activate macrophages into a proinflammatory profile -at least for a short time- early on during infection. In line with this, Saha

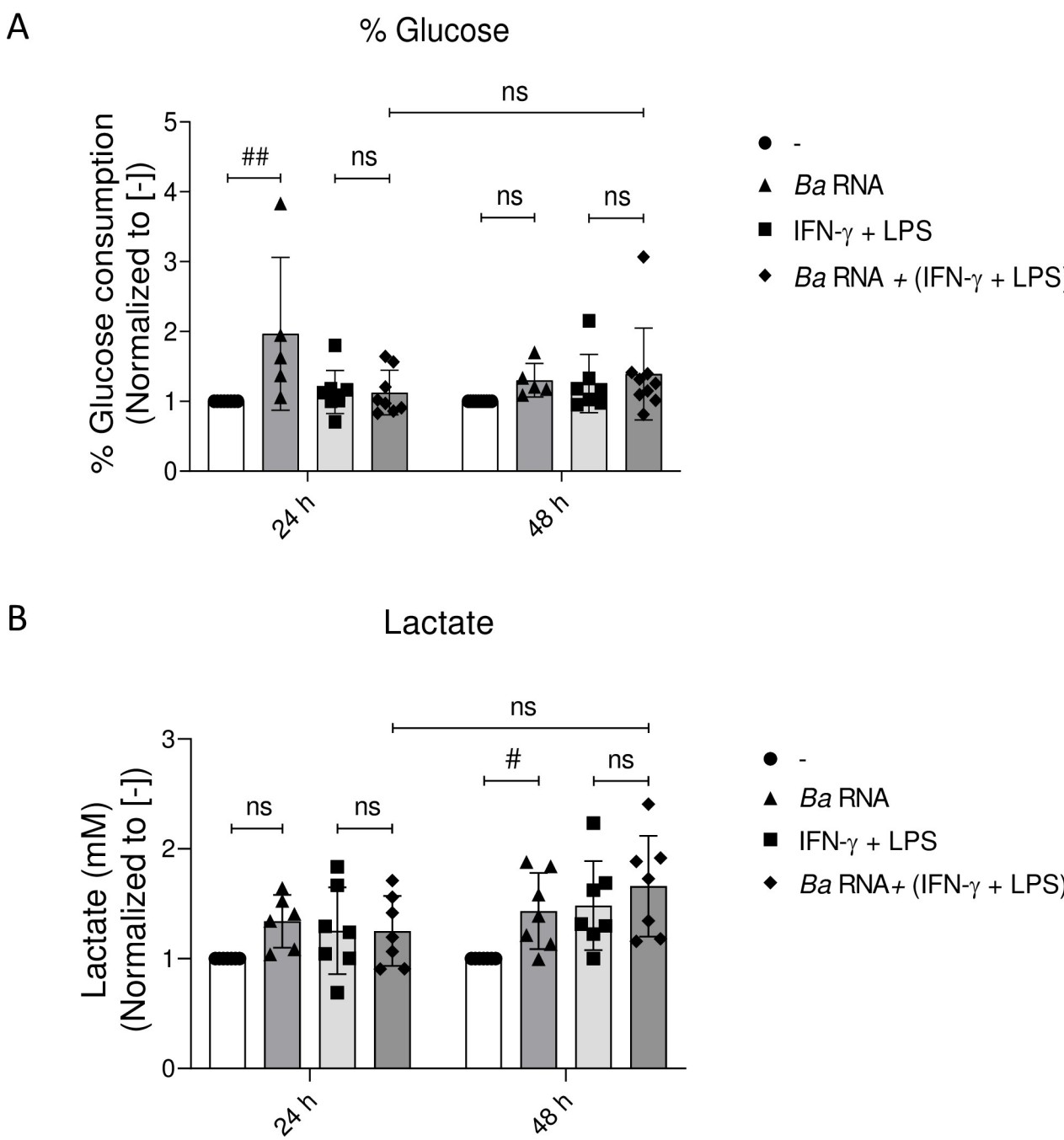

**Fig 8. *Ba* RNA does not modulate the glucose concentration and lactate production.** Human M0 macrophages were stimulated with *Ba* RNA for 24 and 48 h in the presence of IFN-γ + LPS. Supernatants were harvested and (A) glucose concentration and (B) lactate production were determined by a colorimetric assay. Cells treated with IFN-γ and LPS were used as a positive control. Bars indicate the geometric means normalized to untreated cells ± SEM of at least five independent experiments. ns, non-significant. #P<0.05; ##P<0.01 vs. untreated cells (-).

*et al.* demonstrated that TLR7/8 signalling during Hepatitis C infection promotes the differentiation of monocytes and polarization of macrophages to M2, although with a combined M1 and M2 pattern of cytokine secretion [13]. Our results establish that *Ba* RNA activates but does not polarize macrophages. This makes sense if we consider that *Brucella* spp. infection does not generate an inflammatory pathology as massive as other microorganisms [31].

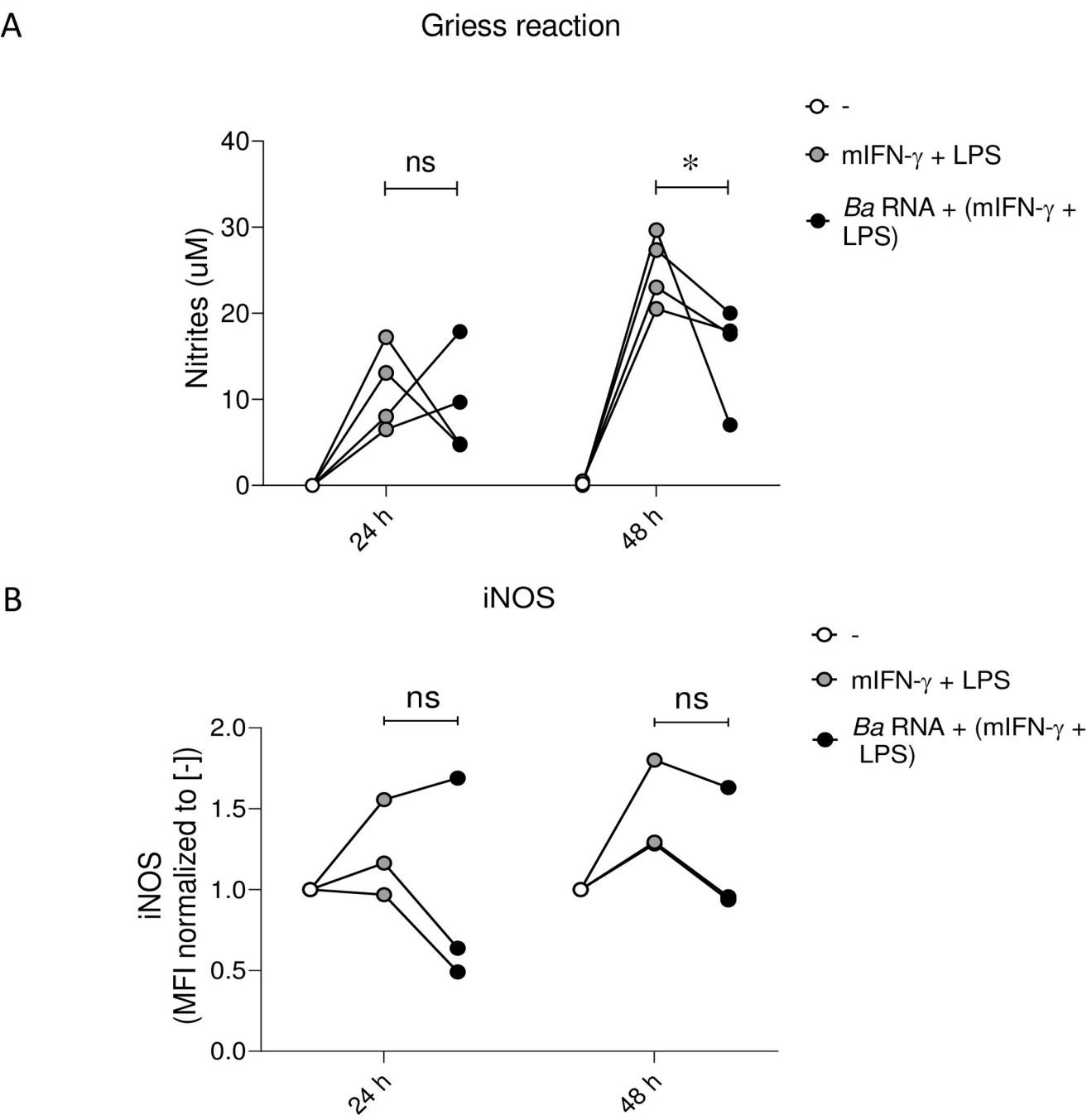

**Fig 9. *Ba* RNA decreases NRS production under M1 conditions at 48 h.** Murine bone marrow macrophages (BMM) were stimulated with *Ba* RNA for 24 and 48 h in the presence of murine IFN-γ + LPS. (A) Afterwards, nitrites were determined in supernatants by Griess assay. Dots indicate the geometric means of four independent experiments. (B) iNOS expression was determined by intracellular staining of treated BMM. Dots indicate the geometric means normalized to untreated cells of three independent experiments. Cells treated with mIFN-γ and LPS were used as a positive control. MFI, mean fluorescence intensity. ns: non-significant; *P<0.05 vs. cells treated with mIFN-γ + LPS.

Hence, *Ba* RNA as a PAMP is not as potent as other immunomodulatory PAMPs such as *E. coli* LPS.

In order to control the progress of the disease, two main components of adaptive immunity play significant roles: IFN-γ secreted by CD4[+] and CD8[+] T cells [32,33] and cytotoxic CD8[+] T

cells which eliminate *Brucella* infected cells [34]. Hence, next we performed experiments to evaluate whether *Ba* RNA was capable of modulating or interfering with M1 polarization of M0 macrophages, *i.e.*, in an established Th1 environment. At 24 h, *Ba* RNA diminished M1-induced CD64 expression and at 48 h, *Ba* RNA could also down modulate M1 induced-CD64 and MHC-II expressions. This, however, was not accompanied by augmented M2 makers. Cytokine secretion was not altered in M1 macrophages treated with *Ba* RNA despite finding a tendency towards an anti-inflammatory cytokine context. In literature, IL-8 secretion could be increased in M1 as well as in M2 macrophages [35]. Also, nowadays, there are much more than M1 and M2 macrophages [2,36] and it is being discovered a wide spectrum between M1 and M2 [37].

We have already shown that *Ba*-infected or *Ba* RNA-treated macrophages show a diminished antigen presentation capacity to CD4$^+$ T cells [8,9,12]. In addition, we and others have demonstrated that phagocytic capacity of *Ba*-infected monocytes is also altered [25,38]. This was also supported in this study, since *Ba* RNA diminishes CD64 expression (type I receptor for the Fc portion of IgG, FcγRI). Therefore, this *vita*-PAMP may be another component besides lipoproteins of *Ba* to contribute to change key microbicidal macrophage actions.

Therefore, our results indicate that *Ba* RNA does not polarize macrophages to a particular profile, but 'interferes' with M1 polarization instead, at least phenotypically.

This kind of M1 interference has already been seen in other infectious pathologies: *Mycobacterium tuberculosis* inhibits transcription of IFN-γ target genes and contributes to the development of the typical pleural effusion of tuberculosis patients [39]. However, granulomatose macrophages express both pro- and anti-inflammatory markers, indicating that polarization is a kind of spectrum more than binary. Also, in *Salmonella typhimurium* infections, *e.g.*, diverse activation status of macrophages between M1 and M2 have been observed [40]. Regarding *Brucella* spp. *in vivo*, there are no determinant results according to macrophage polarization so far. Monocytes from brucellosis patients are not able to polarize to M1 and M2 defined profiles [24].

On the other hand, *Ba* RNA-treated macrophages diminished the IFN-γ-induced production of NRS at 48 h. In concordance with this, Hu *et al.* show that infection of murine macrophages with *Ba* down modulates TXNIP (thioredoxin-interacting protein) expression which promotes survival of the bacteria in those cells through reduction of NRS and Reactive Oxygen Species [41]. Moreover, Kerrinnes *et al.* showed that alternatively activated macrophages infected with *Ba* shifted the production of NRS towards polyamines, favouring the establishment of a chronic infection in mice [42]. We saw no effects on glucose consumption and lactate production on M1 macrophages by *Ba* RNA compared to untreated M1 cells, except for an augmented -although non-significant- lactate production at 48 h in *Ba* RNA-treated M1 macrophages. Czyz *et al.* demonstrated that *Ba* infected macrophages display a metabolic pathway characterized by an augmented lactate production that acts as the only source of carbon and energy [43]. Some *Brucella* virulence factors modulate the host metabolism as reviewed in [44], perhaps it is not the case for *Ba* RNA, or at least, not in this model.

An issue that merits discussion is the differences observed between the two experimental times points evaluated. In our first set of experiments, the results show that *Ba* RNA induces the secretion of pro-inflammatory cytokines (IL-1β and TNF-α) at 24 h and the anti-inflammatory cytokine IL-10 at 24 h and 48 h. However, in our second set of experiments under polarizing conditions, *Ba* RNA interferes with the expression of M1 markers, and it decreases NRS production at 48 h. These results are in line with those previously observed in our laboratory in which at early post-stimulation time points (24 h) either infection with *Ba*, its RNA or synthetic TLR8 agonists activate immune parameters of dendritic cells (DCs) or monocytes. Particularly, we previously published results showing that *Ba* infection induces DCs

maturation, as evidenced by the up-regulation of CD86, CD80, CCR7, CD83, MHC-II, MHC-I and CD40 at 24 h post-infection [45]. Several years later, we also demonstrated that *Ba* RNA induces MHC-II expression on DCs at 24 h post-stimulation [12]. Moreover, we have previously published results demonstrating that the stimulation of human monocytes with *Ba* RNA or synthetic TLR8 agonists for 24 h induce the secretion of the pro-inflammatory cytokines TNF-α and IL-1β [11]. On the other hand, we previously demonstrated that at later post-stimulation time points (48 h), either infection with *Ba* or stimulation with its RNA inhibit the expression of MHC-II and MHC-I in monocytes/macrophages and the consequent antigen presentation to CD4+ and CD8+ T cells, respectively [6,9,11,12].

The apparent discrepancies between the up-regulation of MHC-I and MHC-II in DCs and the downregulation of both molecules in monocytes/macrophages could be explain in terms of the kinetics of *Ba* infection. One explanation is that activation of DCs with *Ba* is likely relevant at the onset of immune response, when Th1 and T CD8+ responses are triggered. At later time points, *Ba* might be able to circumvent these responses to establish a chronic infection by means of different evasion mechanism such as the down-modulation of MHC-I and MHC-II [33] and the consequent interference with M1 polarization of macrophages, where it dwells.

Overall, the results of this study shed light into a relevant aspect whereby *Ba* evades immune host surveillance: the interference with M1 macrophage polarization. Our findings demonstrate that the *vita*-PAMP *Ba* RNA constitute a novel virulence factor whereby this bacterium interferes with M1 polarization of human macrophages, not only by modification of their phenotype but also with their killing activity, specifically by diminishing the production of nitrogen reactive species. This provides new insights about how *Ba* can persist in the presence of Th1 responses, *i.e.*, in the presence of IFN-γ, establishing a chronic infection.

These results also lay the ground for studying more deeply the modulatory properties of microbial RNA in the context of brucellosis, other intracellular infections, and tumours.

## Limitations

This study has some limitations. We designed a model using purified human monocytes differentiated to macrophages with GM-CSF. Nevertheless, other models include differentiation of monocytes with M-CSF (which we used for BMM). Moreover, we could not perform an *in vivo* analysis with blood samples from brucellosis patients (to purify monocytes and differentiate them with GM-CSF as we did for monocytes from healthy blood donors) due to COVID-19 pandemic and the fact that these patients are sub-diagnosed in Argentina. In future research, we would like to obtain a representative number of these samples and deep our understanding of the pathology. We also should design an *in vivo* model with *Ba* infected mice to obtain monocytes and differentiate them to macrophages and evaluate their profile and functionality. Furthermore, we did not deeply evaluate the mechanisms downstream the signalling pathway of *Ba* RNA-stimulation of M1 macrophages. Evaluation of the receptor/s and metabolic pathways of M1-interefered macrophages could be interesting studies to make a better understanding of the disease. Finally, when understanding the intracellular pathway triggered by *Ba* RNA, we will be able to propose molecules or drugs to revert the interference in M1 macrophages and provide perhaps some new treatments to be tested for human and animal brucellosis.

## Supporting information

**S1 Fig. *Ba* RNA does not induce loss of cell viability.** Monocytes derived from peripheral blood were differentiated to macrophages with GM-CSF for 5–7 days and then stimulated with *Ba* RNA (5 μg/ml) for (A) 24 and (B) 48 h. They were then stained with Annexin V-FITC

and Propidium Iodide (IP) and then analyzed to evaluate early apoptosis (Annexin V$^+$/IP$^-$), late apoptosis (Annexin V$^+$/IP$^+$) or necrosis (Annexin V$^-$/IP$^+$). Cells treated with Paraformaldehyde (PFA) were used as a positive control for late apoptosis. $^{***}$P$<$0.001 vs. untreated cells (-).
(TIF)

**S2 Fig. *Ba* RNA does not induce loss of cell viability during polarization to M1.** Monocytes derived from peripheral blood were differentiated to macrophages with GM-CSF for 5–7 days and then stimulated with *Ba* RNA (5 μg/ml) for (A) 24 and (B) 48 h, in the presence of IFN-γ + LPS. They were then stained with Annexin V-FITC and Propidium Iodide (IP) and then analyzed to evaluate early apoptosis (Annexin V$^+$/IP$^-$), late apoptosis (Annexin V$^+$/IP$^+$) or necrosis (Annexin V$^-$/IP$^+$). Cells treated with Paraformaldehyde (PFA) were used as a positive control for late apoptosis. $^{***}$P$<$0.001 vs. untreated cells (-).
(TIF)

**S3 Fig. *Ba* RNA does not alter the pro/anti-inflammatory cytokine profile of M1 macrophages.** Monocytes derived from peripheral blood were differentiated to macrophages with GM-CSF for 5–7 days and then stimulated with *Ba* RNA (5 μg/ml) for 24 and 48 h in the presence of IFN-γ + LPS. Secretion of TNF-α, IL-1β and IL-10 was quantified in culture supernatants. Afterwards, the ratio of pro/anti-inflammatory cytokines was calculated for each treatment at each time point. Dots indicate the ratio of five independent experiments. ns, non-significant. $^*$P$<$0.05; $^{***}$P$<$0.001 vs. untreated cells (-).
(TIF)

**S4 Fig. *Ba* RNA down-modulates M1-induced mMHC-II surface expression.** Murine BMM were stimulated with *Ba* RNA (5 μg/ml) for 24 and 48 h in the presence of mIFN-γ + LPS. Afterwards, mMHC-II surface expression was assessed by flow cytometry. Dots indicate the geometric means of five independent experiments. ns, non-significant; $^{***}$P$<$0.001 vs. untreated cells (-) or mIFN-γ + LPS.
(TIF)

## Acknowledgments

We would like to thank the staff of ANLIS-Malbrán (Buenos Aires) for allowing us to use the Biosafety Level 3 Laboratory facilities to prepare cell cultures of *Ba*. We also thank the technical assistance with flow cytometry assays provided by Alejandro Benatar.

## Author Contributions

**Formal analysis:** Paula Barrionuevo.

**Investigation:** Agustina Serafino, José L. Marin Franco, Mariano Maio, Aldana Trotta, Melanie Genoula, Luis A. Castillo, Federico Birnberg Weiss, José R. Pittaluga, M. Ayelén Milillo.

**Methodology:** Agustina Serafino, José L. Marin Franco, Mariano Maio, M. Ayelén Milillo.

**Resources:** Luciana Balboa, Paula Barrionuevo.

**Supervision:** Luis A. Castillo, Luciana Balboa, Paula Barrionuevo, M. Ayelén Milillo.

**Validation:** José L. Marin Franco, Melanie Genoula, Paula Barrionuevo.

**Visualization:** Agustina Serafino.

**Writing – original draft:** Agustina Serafino, Paula Barrionuevo, M. Ayelén Milillo.

**Writing – review & editing:** Paula Barrionuevo, M. Ayelén Milillo.

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
