## [Decision Letter · Decision Letter 0]

17 Jul 2022

Dear Aileen Milillo,

Thank you very much for submitting your manuscript "Brucella abortus RNA interferes with the M1 polarization of human macrophages" for consideration at PLOS Neglected Tropical Diseases. As with all papers reviewed by the journal, your manuscript was reviewed by members of the editorial board and by several independent reviewers. In light of the reviews (below this email), we would like to invite the resubmission of a significantly-revised version that takes into account the reviewers' comments. 

We cannot make any decision about publication until we have seen the revised manuscript and your response to the reviewers' comments. Your revised manuscript is also likely to be sent to reviewers for further evaluation.

Sincerely,

Ruifu Yang

Section Editor

Ruifu Yang

Section Editor

Reviewer's Responses to Questions

**Key Review Criteria Required for Acceptance?**

**Methods**

-Are the objectives of the study clearly articulated with a clear testable hypothesis stated?

-Is the study design appropriate to address the stated objectives?

-Is the population clearly described and appropriate for the hypothesis being tested?

-Is the sample size sufficient to ensure adequate power to address the hypothesis being tested?

-Were correct statistical analysis used to support conclusions?

-Are there concerns about ethical or regulatory requirements being met?

Reviewer #1: -Are the objectives of the study clearly articulated with a clear testable hypothesis stated?

Yes

-Is the study design appropriate to address the stated objectives?

Yes

-Is the population clearly described and appropriate for the hypothesis being tested?

Yes

-Is the sample size sufficient to ensure adequate power to address the hypothesis being tested?

Not applicable.

-Were correct statistical analysis used to support conclusions?

Yes.

-Are there concerns about ethical or regulatory requirements being met?

No.

Reviewer #2: The manuscript is methodologically sound.

Reviewer #3: The objectives and the study design are congruent with the objectives, but the hypothesys must be enunciated to the reader. Methods are correct, but please indicate in more detail the flow cytometric analysis: it is not clear if the normalization is performed with percentages or MFI. In addition, please indicate the formula used for normalization.

**Results**

-Does the analysis presented match the analysis plan?

-Are the results clearly and completely presented?

-Are the figures (Tables, Images) of sufficient quality for clarity?

Reviewer #1: -Does the analysis presented match the analysis plan?

Yes.

-Are the results clearly and completely presented?

Yes.

-Are the figures (Tables, Images) of sufficient quality for clarity?

Figure resolution need improvement.

Reviewer #2: The results are properly presented and the figures have sufficient quality.

Reviewer #3: This section is correct, although writing must be improved, since its written in a very informal English. The results are clearly described, but considering the modifications suggested, it must be improved. Regarding the figures, please improve them by indicating, for example in fig. 1, if “normalization to negative” indicates negative control (isotype control) or media condition. It could be useful also to draw a line though the replicate to track the changes as they occur (Media – Ba RNA – IFNg+LPS).

**Conclusions**

-Are the conclusions supported by the data presented?

-Are the limitations of analysis clearly described?

-Do the authors discuss how these data can be helpful to advance our understanding of the topic under study?

-Is public health relevance addressed?

Reviewer #1: -Are the conclusions supported by the data presented?

Yes.

-Are the limitations of analysis clearly described?

No.

-Do the authors discuss how these data can be helpful to advance our understanding of the topic under study?

Not exactly.

-Is public health relevance addressed?

No.

Reviewer #2: A significant part of the manuscript are negative results and, because of this, the main conclusion is that B. abortus RNA activates macrophages and phenotypically inhibits polarization of macrophages under in vitro M1 polarization conditions. Some controls are necessary to properly conclude this (see summary and general comments).

Reviewer #3: Conclusions are supported by the data. Nevertheless, discussion must be improved. i) limitations are not addressed; ii) Public relevance is not highlighted; iii) an in-depth analysis of the observations is not adopted. Please improve this section comprehensively. Discuss the observations regarding “phenotypic changes”: supported by the literature, discuss the implications in cell function in the context of Brucella abortus infection. Please discuss how these data can be helpful to advance our understanding of the topic under study.

**Editorial and Data Presentation Modifications?**

Reviewer #1: Minor Revision

Reviewer #2: (No Response)

Reviewer #3: (No Response)

**Summary and General Comments**

Reviewer #1: Brucella abortus (Ba) is an intracellular pathogen which survives in monocytes and macrophages. This manuscript report their findings about the mechanism about Brucella survival in macrophages, which is an interesting field about the interaction between Brucella and host cells. In their previous study, the authors have shown Ba RNA stimulate pro-inflammatory cytokine of monocytes. In this study, they test whether Ba RNA polarize macrophages. Their results showed that Ba RNA diminished t CD64, and MHC-II surface expression on macrophages at 48h, but did not modify CD206, DC-SIGN and CD163 surface expression, indicating that Ba RNA interfere M1 but not M2. Further analysis showed that Ba RNA did not modify M1- glucose consumption or lactate production. but production of Nitrogen Reactive Species diminished in Ba RNA-treated M1 macrophages. All these results show that Ba RNA could alter the proper immune response set to counterattack the bacteria. This work also demonstrates that RNA could also functions as a virulence factor of Brucella.

Specific comments

1, The main findings of this study is that Ba RNA does not activate both M1 and M2 polarization, but interfere M1 polarization of macrophages. This finding should be made more clear both in the title and abstract.

2, Two time points, 24 and 48h, were checked for surface expression of cytokines and biomakers. No explain or discussion about the expression difference between the two time points, particularly from the bacteria host interaction aspects.

3, What effects of the macrophage polarization will be generated on Ba survival or immune response to the bacteria? This should be discussed according to the results.

Reviewer #2: The paper describes that the RNA from B. abortus induces the activation of macrophages but is not capable of polarizing them neither to M1 nor to M2. The authors showed that B. abortus RNA blocks the differentiation to M1 (induced by treatment with LPS and IFN-gamma) but this blockage is not the result of inducing them to express M2 markers. The authors also show that RNA treatment of macrophages induces glucose consumption but not lactate production and a mild reduction (only at 48 hrs post-treatment) of NO production in macrophages although the production of iNOS was not significantly modified.

Major concerns

1- The main conclusion of the manuscript is that RNA from B. abortus is not capable of polarizing macrophages (to M1 or M2) but activates them. Additionally, if the macrophages are polarized to M1 in the presence of RNA, this polarization is phenotypically inhibited. Most of the results presented are negative. This is: no polarization was observed when the macrophages were treated with the RNA (Figures 1 and 3), partial macrophage activation was seen (IL-1 beta only at 24 hrs, TNF-alpha only at 48 hrs but not IL-8, Figure 2), it does not interfere with the secretion of pro-inflammatory cytokines (Figure 6), it does not modulate M2 markers when inhibition of M1 polarization is inhibited (Figure 7) and it does not modulate glucose concentration and lactate production (Figure 8). The main biological conclusion is that the RNA could be inhibiting M1 polarization and, in this way, modulating the immune response during infection but to conclude this a central control should be done (see point below).

2- The authors conclude that B. abortus RNA activates macrophages, but a control should be included to show that the RNA preparations are not contaminated with other molecules that could potentially activate macrophages. Have the authors treated the RNA with RNAse and see if the effect is lost? This control should also be included in the inhibition of M1 polarization, glucose consumption and NO production.

Reviewer #3: The manuscript from Serafino et. al constitutes an interesting and valuable input to the field of Brucella abortus immunology. However, this Reviewer consider that it must be improved to be published by Plos Pathogens Journal. 

The Major Revision is requested because this reviewer considers that language must be improved (it is written in a very informal English and the relevant observations are not highlighted), but also additional data analysis must be done to improve the manuscript. Therefore, please calculate the RATIO between TNF-a/IL10 and / or IL1b/IL10 to assess changes in the pro-inflammatory/anti- inflammatory balance induced by Ba RNA. Also, glucose analogue incorporation could be performed to clarify glucose consumption, since de assessment of glucose in culture supernatants presents high variability.

Finally, regarding mouse macrophages: this reviewer considers that this set of experiments does not fit with the rest of the manuscript. Therefore, please take out the figure OR perform the same experiments with HUMAN macrophages.

PLOS authors have the option to publish the peer review history of their article (what does this mean?). If published, this will include your full peer review and any attached files.

Reviewer #1: No

Reviewer #2: No

Reviewer #3: Yes: Maria Florencia Quiroga
---

## [Decision Letter · Decision Letter 1]

20 Oct 2022

Dear M. Aileen Milillo,

Thank you very much for submitting your manuscript "Brucella abortus RNA does not polarize macrophages to a particular profile but interferes with M1 polarization" for consideration at PLOS Neglected Tropical Diseases. As with all papers reviewed by the journal, your manuscript was reviewed by members of the editorial board and by several independent reviewers. In light of the reviews (below this email), we would like to invite the resubmission of a significantly-revised version that takes into account the reviewers' comments. 

We cannot make any decision about publication until we have seen the revised manuscript and your response to the reviewers' comments. Your revised manuscript is also likely to be sent to reviewers for further evaluation.

Sincerely,

Ruifu Yang

Section Editor

Ruifu Yang

Section Editor

Reviewer's Responses to Questions

**Key Review Criteria Required for Acceptance?**

**Methods**

-Are the objectives of the study clearly articulated with a clear testable hypothesis stated?

-Is the study design appropriate to address the stated objectives?

-Is the population clearly described and appropriate for the hypothesis being tested?

-Is the sample size sufficient to ensure adequate power to address the hypothesis being tested?

-Were correct statistical analysis used to support conclusions?

-Are there concerns about ethical or regulatory requirements being met?

Reviewer #1: -Are the objectives of the study clearly articulated with a clear testable hypothesis stated?

Yes

-Is the study design appropriate to address the stated objectives?

Yes

-Is the population clearly described and appropriate for the hypothesis being tested?

Yes

-Is the sample size sufficient to ensure adequate power to address the hypothesis being tested?

Not applicable.

-Were correct statistical analysis used to support conclusions?

Yes.

-Are there concerns about ethical or regulatory requirements being met?

No.

Reviewer #2: (No Response)

Reviewer #3: (No Response)

**Results**

-Does the analysis presented match the analysis plan?

-Are the results clearly and completely presented?

-Are the figures (Tables, Images) of sufficient quality for clarity?

Reviewer #1: -Does the analysis presented match the analysis plan?

Yes.

-Are the results clearly and completely presented?

Yes.

-Are the figures (Tables, Images) of sufficient quality for clarity?

Yes.

Reviewer #2: (No Response)

Reviewer #3: (No Response)

**Conclusions**

-Are the conclusions supported by the data presented?

-Are the limitations of analysis clearly described?

-Do the authors discuss how these data can be helpful to advance our understanding of the topic under study?

-Is public health relevance addressed?

Reviewer #1: -Are the conclusions supported by the data presented?

Yes.

-Are the limitations of analysis clearly described?

No.

-Do the authors discuss how these data can be helpful to advance our understanding of the topic under study?

Yes.

-Is public health relevance addressed?

yes

Reviewer #2: (No Response)

Reviewer #3: (No Response)

**Editorial and Data Presentation Modifications?**

Reviewer #1: (No Response)

Reviewer #2: (No Response)

Reviewer #3: (No Response)

**Summary and General Comments**

Reviewer #1: (No Response)

Reviewer #2: (No Response)

Reviewer #3: The authors have addressed succesfully all the issues raised by this reviewer. As it is now, the manuscript is suitable for publication.

PLOS authors have the option to publish the peer review history of their article (what does this mean?). If published, this will include your full peer review and any attached files.

Reviewer #1: No

Reviewer #2: No

Reviewer #3: No
---

## [Editor Report · Decision Letter 2]

14 Nov 2022

Dear Dr.
M. Ayelén Milillo,

We are pleased to inform you that your manuscript 'Brucella abortus RNA does not polarize macrophages to a particular profile but interferes with M1 polarization' has been provisionally accepted for publication in PLOS Neglected Tropical Diseases.

Best regards,

Ruifu Yang

Section Editor

Ruifu Yang

Section Editor

---

## [Editor Report · Acceptance letter]

24 Nov 2022

Dear Dr Milillo,

We are delighted to inform you that your manuscript, "</i>Brucella abortus</i> RNA does not polarize macrophages to a particular profile but interferes with M1 polarization," has been formally accepted for publication in PLOS Neglected Tropical Diseases.

Best regards,

Shaden Kamhawi

co-Editor-in-Chief

Paul Brindley

co-Editor-in-Chief
